# Inhibition of *Listeria monocytogenes* Cocktail Culture Biofilms on Crab and Shrimp Coupons and the Expression of Biofilm-Related Genes

**DOI:** 10.3390/antibiotics12061008

**Published:** 2023-06-04

**Authors:** Pantu Kumar Roy, So Hee Kim, Eun Bi Jeon, Eun Hee Park, Shin Young Park

**Affiliations:** Institute of Marine Industry, Department of Seafood Science and Technology, Gyeongsang National University, Tongyeong 53064, Republic of Korea; vetpantu88@gmail.com (P.K.R.); thgml9903@naver.com (S.H.K.); eunb61@naver.com (E.B.J.); pehbeda@hanmail.net (E.H.P.)

**Keywords:** *Listeria monocytogenes*, quercetin, biofilm, crabs, shrimps, food safety, relative expression

## Abstract

*Listeria monocytogenes*, a bacterium that is transmitted by tainted food, causes the infection listeriosis. In this study, quercetin was tested for its antibacterial properties and effectiveness as a food additive in preventing the growth of *L. monocytogenes* cocktail (ATCC19117, ATCC19113, and ATCC15313) biofilms on crabs and shrimps. Quercetin showed the least bactericidal activity and no discernible microbial growth at a minimum inhibitory concentration (MIC) of 250 µg/mL. The biofilm inhibition was performed at sub-MICs (1/2, 1/4, and 1/8 MIC). There was no quercetin added to the control group. Additionally, the present work examines the expression of various genes related to biofilm formation and quorum sensing (*flaA, fbp, agrA, hlyA, and prfA*). The levels of target genes were all significantly down-regulated. Quercetin (0–125 µg/mL) on the surfaces of the crab and shrimp was studied; its inhibitory effects were measured as log reductions at 0.39–2.31 log CFU/cm^2^ and 0.42–2.36 log CFU/cm^2^, respectively (*p* < 0.05). Quercetin reduced the formation of biofilms by disrupting cell-to-cell connections and causing cell lysis, which led to the deformation of the cells, evidenced by FE-SEM (field-emission scanning electron microscopy). These findings emphasize the significance of using natural food agents to target bacteria throughout the entire food production process.

## 1. Introduction

Consuming contaminated food can result in foodborne illnesses, and their rising incidence is a critical public health concern [1]. *L. monocytogenes*, a ubiquitous bacterium, can develop biofilm on foods and food-contact surfaces, aiding its survival in environments where food is processed [2]. It is unknown what the precise molecular basis underlying the ability to form biofilms in *L. monocytogenes* is because non-motile mutants are unable to do so [3]. In quorum-sensing (QS), *agr* and *PrfA* play vital roles in the formation of biofilms [4,5]. Biofilms exhibit a 10-fold greater resistance to all antimicrobial agents than planktonic cells. This makes it challenging to eliminate biofilms with unvarying antibiotics and washing supplies [6]. Initial adhesion, microcolony generation, and biofilm progress are the phases of biofilm development, as shown by microscopic investigations [7]. In order to lessen the detrimental effects of biofilms, chemicals including sodium hypochlorite or sodium hydroxide are widely used for food processing [8,9]. By corroding tools and materials, these techniques can harm the environment [10]. Therefore, developing an effective method that can control and eliminate biofilm is crucial. Biofilms are bacterial growths that defend themselves by implanting cells in extracellular polymeric substances (EPS) [11,12]. This makes the microbes better equipped to endure coming into contact with antimicrobial chemicals [13,14]. The production of EPS, resource capture, detachment, and dispersal are regulated by a number of genes that are connected to biofilms [15]. Many diseases in humans may originate from contaminated surfaces. Permanently attached to both biotic and abiotic surfaces, biofilms are bacterial populations that have incorporated an extracellular polymeric matrix (ECM) into their own structure [16].

Food and food-contact-surface biofilms are responsible for food product quality, quantity, and safety [17,18]. They also damage surfaces and equipment and regularly infect food, constituting a serious threat; controlling these types of biofilms is a major challenge in the food industry [11,19]. Plant-extracted antibacterial substances are often regarded as safe, effective, and environmentally friendly for the prevention of foodborne illnesses [20,21,22]. Natural plant extracts are used extensively for food safety and illness prevention [23]. Plants can be used as a different treatment for *L. monocytogenes* biofilms [24,25]. Through a variety of mechanisms, phenolic compounds, in particular, stop this pathogen from forming biofilms. For instance, gallic acid treatment (at 5.8 mM) reduced *L. monocytogenes* biofilms [26]. These modifications were related to a reduction in biofilms on polystyrene surfaces. In a similar manner, resveratrol (0.2–0.8 mM) decreased the overall biomass of *L. monocytogenes* biofilms and reduced the metabolic movement of the adhering cells [27,28].

Preventive measures are required for refining safety and food quality by using inhibitory chemicals that interfere with QS [27]. Modulating virulence, extracellular enzymes, the formation of biofilms, subordinate metabolites, and motility are only some of the characteristics that QS controls in bacteria. These characteristics often lead to food spoilage, rendering it unsafe for human consumption. Therefore, scientists have been trying to find ways to utilize inhibitory substances to cut off this line of communication and make food safer and better. Numerous scientific investigations have demonstrated that phenolic-rich plant organic extracts can inhibit QS in a range of bacterial species. In order to reproduce, grow, and defend themselves against pathogens, plants rely on a wide range of chemical substances. Natural plant extracts are classified as phenolic acids, lignans, stilbenes, and flavonoids. QS enables cell communication and helps germs to endure harsh circumstances [29]. These molecules are produced when bacteria exceed a preset absorption threshold, which controls the expression of virulence genes [30]. QS controls the amalgamation of numerous virulence factors as well as the motility and formation of biofilms [29]. Numerous bacterial QS can be hampered by the phenolic compounds produced by plants [27]. Flavonoids are hydroxylated phenolic composites that garner a lot of attention for their prospective health benefits to humans in the forms of functional meals and new therapies. They are the most well-known classes of natural products, and they are abundant in the foods that people eat every day. The hydroxylated conjugates of several naturally occurring flavonoids exhibit a wide range of physiological actions. Antioxidant power is a hallmark of flavonoids. The function of flavonoids and their metabolites in biological functioning has been the subject of numerous studies. These studies advocate for the screening of numerous flavonoids to establish their usefulness to humanity. Flavonoids, such as quercetin, kaempferol, and others, are linked to a lower risk of emerging chronic diseases including diabetes, cardiovascular disease, atherosclerosis, stroke, and microbial infections, according to epidemiological research.

A total of 80 percent of persistent bacterial illnesses are linked to biofilms. Since many persistent infections originate in biofilms, it becomes increasingly challenging to eradicate these germs. The discovery and progress of new pharmaceuticals with enhanced beneficial action have been the subjects of intensive study in recent years, with much attention paid to new methodologies and technical breakthroughs.

Because of the rise in drug-resistant organisms, the flavonoid quercetin has emerged as a promising new treatment option [31]. Quercetin is widely used for its antioxidant, free radical scavenging, anticancer, and neuroprotective properties, and it is found in many different types of ethnic plants. Quercetin is traditionally used to treat infectious disorders, and the effectiveness of different forms of the compound depends on its structural makeup.

Microbial biotransformation is a relatively new method of production. The sustainable production of flavonoids is facilitated by microbial biotransformation, which involves the modification of microbes in actual microbial cell factories. With the added benefit of mass synthesis, the selectivity of naturally occurring and synthetically produced flavonoids can be enhanced by this potent method. Novel medications are in high demand due to the promise of microbial biotransformation for the discovery and manufacturing of flavonoids, which can treat a wide range of pathological illnesses at a low cost while protecting the environment from harmful waste. Alternative plant-based substances can be used to regulate the biofilms of *L. monocytogenes*, and quercetin is a well-studied flavonoid molecule with useful properties in this regard. In addition to their potential ability to suppress the QS system, flavonoids are now used for antibacterial, antioxidant, anti-inflammatory, and anticancer activities [32,33]. Quercetin, a flavonoid-based substance, is found in a variety of fruits and vegetables (apples, tomatoes, onions, berries, red grapes, and tea [33]) and has five hydroxyl groups with a three-ring structure [34,35]. Antioxidants can lower oxidative stress and decrease the development of biofilms by removing reactive oxygen species (ROS) [36]. Antioxidants are therefore effective antibiofilm agents [37]. As a means of survival, oxidative stress is one of the main mechanisms of biofilm formation. Quercetin has also been shown to have antibacterial activity against *Staphylococcus aureus* [38], *Escherichia coli* [39], and *Pseudomonas aeruginosa* [40]. The ability of *L. monocytogenes* to adhere to stainless steel surfaces has been shown to be reduced by exposure to the flavonoids and the phenolic acids gallic, ferulic, and caffeic acids [41].

Therefore, we sought to evaluate the quercetin effects at sub-MIC to inhibit the formation of *L. monocytogenes* cocktail culture biofilms on crabs and shrimps, as well as the relative expression of different genes.

## 2. Results

### 2.1. MIC Determination

As determined by the MIC test, the MICs for the *L. monocytogenes* cocktail culture were 125 μg/mL of quercetin. A sub-MIC is a concentration that is inactive for microbial development but active in inhibiting microbial pathogenicity [24]. This investigation revealed that quercetin inhibits virulence and QS gene expression at sub-MICs (125 μg/mL).

### 2.2. Effects of Quercetin against L. monocytogenes Cocktail Culture Biofilm on Foods

Biofilms were formed on the surfaces of the crabs and shrimps at various quercetin concentrations, and the inhibitory effects are shown in Figure 1 and Figure 2. The *L. monocytogenes* cocktail culture biofilm reduction on the crabs was 0.39, 1.25, and 2.31 log CFU/cm^2^_,_ and on the shrimps, it was 0.42, 1.27, and 2.36 log CFU/cm^2^ at 1/8, 1/4, and 1/2 MIC, respectively. The inhibitory effects were significantly different at 1/2 MIC for quercetin compared to the other group (*p* < 0.05). Biofilm formation was greater on crabs, and inhibition was lower on the crabs than on the shrimps’ surface (Figure 1 and Figure 2). Protective bacterial growth occurred environmentally on the crabs and formed one of the key sources of listeriosis, so it should not be contaminated with *L. monocytogenes* bacteria throughout food processing [24].

### 2.3. Visual Analysis of Biofilm

Quercetin has antibacterial properties through membrane permeation and oxidant scavenging activities. Further, it can inhibit the formation of biofilms related to ROS. The effect of quercetin against the *L. monocytogenes* cocktail culture biofilms on the crabs and shrimps was investigated by FE-SEM (Figure 3). In particular, an extensive biofilm was formed on the surface of the crabs. It has been reported that *L. monocytogenes* biofilms are inhibited by quercetin, which is important for this study [24]. It was noteworthy that biofilm inhibition was much higher at 1/2 MIC than at 1/8 MIC. Additionally, the inhibitory effects of quercetin on the *L. monocytogenes* cocktail culture biofilm were visually confirmed.

### 2.4. Relative Expression of Gene

The different sub-MICs of quercetin were used for RT-PCR to assess the different primer expressions of pathogenicity and the QS factor. With the exception of 1/8 MIC, primer expression was significantly suppressed (*p* < 0.05) under the different sub-MIC levels of quercetin in comparison to the other groups (Figure 4).

## 3. Materials and Methods

### 3.1. Bacterial Strains and Culture

This investigation used three *L. monocytogenes* (ATCC 15313, ATCC 19115, and ATCC 1917) strains. In a 50 mL conical tube (SPL Life Sciences Co., Ltd., Gyeonggi-do, Republic of Korea), 100 µL of aliquot cultures (10^8^ to 10^9^ CFU/mL) was inoculated into 10 mL of tryptic soy broth (TSB; BD Difco, Sparks, NV, USA) at 30 °C for 24 h at 220 rpm in a shaking incubator (VS-8480, Vision Scientific, Gyeonggi-do, Republic of Korea). A portion of the cultured culture (100 µL) was placed in 10 mL TSB for another 24 h. After centrifuging at 10,000× *g* for 10 min at 4 °C, the samples were washed with Dulbecco’s phosphate-buffered saline (DPBS; Sigma-Aldrich, St Louis, MO, USA), and then diluted in peptone water (PW; BD Diagnostics, Franklin Lakes, NJ, USA). The mixed culture was made by mixing equal quantities of each strain. The mixed culture was used to make a suspension of 10^5^ CFU/mL and cultured on PALCAM agar (Oxoid, Hampshire, UK) medium at 30 °C for 24 h.

### 3.2. Quercetin Preparation

Quercetin (Q–4951, Sigma-Aldrich, Gyeonggi-do, Republic of Korea) was obtained and used for our experiments. Quercetin was dissolved in dimethyl sulfoxide (DMSO) to make the stock solution and was stored at 1 mg/mL for further experiments.

### 3.3. MIC Determination

To analyze the planktonic behavior of *L. monocytogenes*, its susceptibility and growing response to quercetin were tested. The MIC of quercetin was calculated using the Clinical & Laboratory Standards Institute (CLSI) microbroth dilution method with a few modifications [42]. In a 96-well microtiter plate, quercetin was diluted in TSB (0 to 500 µL/mL) with 100 µL of suspension (Corning, Inc., Corning, NY, USA). Each bacterium’s immediate culture was additionally diluted to achieve the required final concentration (10^5^ CFU/mL) and then incubated at 30 °C for 24 h. The MIC was used to identify the lowest concentrations of quercetin that suppressed the growth of the bacteria visibly. Sub-MICs were calculated by observing the growth of bacteria over a 24 h period and the absorbance was checked using a microplate reader. Quercetin at 250 µg/mL was our MIC.

### 3.4. Formation and Detachment of Biofilm on Foods

The crab and shrimp surfaces (2 cm, 2 cm, and 0.1 cm, respectively) were treated as previously described [43,44]. To eliminate any leftover substance, oil, and bacteria, the food samples were washed with distilled water (DW) and 70% ethanol. The mixed culture (100 µL) was placed in a 50 mL Falcon tube with a coupon in 10 mL TSB for biofilm growth. At different sub-MIC conditions (0, 1/2, 1/4, and 1/8 MIC), the cells were incubated for 24 h. After biofilm development, the samples were washed with DW to eliminate any detached adherent cells. After that, the samples were homogenized in 710 mL WhirlPak filter stomacher bags with 90 mL of 0.1% PW (Oxoid) using a stomacher (BagMixer 400; Interscience, Paris, France) at a maximum speed for 2 min. Then, the suspensions were cultured on PALCAM agar; cell counts were determined using the plate count method. Log CFU/cm^2^ was used to describe the biofilm cells.

### 3.5. Field Emission Electron Microscope (FE-SEM) for Visual Analysis of Biofilms

This experiment was performed using a FE-SEM (Carl Zeiss, Oberkochen, Germany) method, following previous studies, to confirm the effects of quercetin on food surfaces [11]. Samples were prepared by a series of fixing and dehydrating steps. For prefixation, the samples were kept for 4 h in a 2% glutaraldehyde solution (Sigma, St. Louis, MO, USA) at room temperature. Prefixation was achieved by washing the samples in PBS for 10 min each time. After that, postfixation was carried out for 2 h with a 2% osmium tetroxide solution (Sigma-Aldrich) at room temperature and washed with PBS for 10 min. Then, the samples were successively dehydrated with ethanol solutions (50, 60, 70, 80, and 90% ethanol, respectively). Following that, the samples were treated three times with 100% ethanol. For 15 min, the samples were submerged in ethanol and hexamethyldisilazane (Sigma-Aldrich) in 3:1, 1:1, and 1:3 ratios. Three final treatments with 100% hexamethyldisilazane were carried out. Carbon tape was used to secure the samples to an aluminum stub, and the samples were then coated with palladium gold. The FE-SEM equipment was operated at a 5 kV acceleration voltage and at a working distance of 5 mm.

### 3.6. Expression of Genes Analysis by Real-Time PCR (RT-PCR)

With some minor modifications, the experiment was performed by an earlier procedure [44], RT-PCR was performed to confirm the expression of the genes. The pellet was obtained after culturing the bacteria at 30 °C for 24 h. The total RNA was extracted using the RNeasy Mini Kit (Qiagen, Hilden, Germany). Using a NanoDrop spectrophotometer, the RNA concentrations were assessed at 260/280 and 260/230 nm, respectively (Bio-Tek Instruments, Chicago, IL, USA). A Maxime RT PreMix (Random Primer, iNtRON Biotechnology Co., Ltd., Gyeonggi-do, Republic of Korea) Kit was used for cDNA synthesis. Different primers were used in this study (Table 1). The 16S ribosomal RNA gene was employed as a reference standard. The sample of complementary DNA was combined with the appropriate primers and 20 µL of Power SYBR Green PCR Master Mix (Applied Biosystems, Thermo Fisher Scientific, Warrington, UK). Analysis via RT-PCR was performed using a CFX Real-Time PCR System (Bio-Rad, Hercules, CA, USA). The cDNA (1 µL) was used as a template with a 2X Real-Time PCR Master Mix for RT-qPCR. We used a CFX Real-Time PCR System (Bio-Rad, Hercules, CA, USA) to conduct the real-time PCR analyses. The PCR reaction technique began with an initial denaturation at 95, 50, and 72 °C for 20 s, respectively [45,46]. Following the conclusion of the PCR cycles, the melting curves were calculated to confirm the relative quantification of specific genes.

### 3.7. Statistical Analysis

Every test was performed in three replicates. The results are presented as the mean ± standard error of the mean (SEM). The experimental data were analyzed by ANOVA using SAS software (version 9.2; SAS Institute, Cary, NC, USA). Significance was determined using Duncan’s multiple-range test (*p* < 0.05).

## 4. Discussion

The usual substances derived from plants are proposed to be possibly beneficial to biofilm inhibition. Quercetin is one example of an all-purpose inhibitor. The FDA approved high-purity quercetin as a component in certain food types in 2010 at doses up to 500 mg [42]. The present experiment set out to see if quercetin could stop *L. monocytogenes* mixed culture growth at concentrations of 125 µg/mL. We found that quercetin had a significant bactericidal impact on a cocktail culture of *L. monocytogenes* and inhibited biofilm formation.

The quercetin compound differs widely from species to species. Quercetin was used at 250 µg/mL for *Salmonella* Typhimurium and *P. aeruginosa*, and at 80 µg/mL for *Klebsiella pneumoniae*, 120 µg/mL for *Chromobacterium violaceum*, and 95 µg/mL for *Yersinia enterocolitica* [42,47]. Quantitative assessments of quercetin’s antibacterial activity against *P. aeruginosa*, *Staphylococcus aureus*, and *E. coli* were performed previously. For many strains, higher concentrations significantly raised the minimal inhibitory concentration. The MIC for *S. aureus*, however, was just 100 µg/mL, indicating far greater suppression. Recent research has shown that at sub-minimum inhibitory doses (sub-MICs), quercetin inhibits the growth of *Enterococcus faecalis* MTCC 2729. SEM and confocal laser scanning microscopy (CLSM) images confirmed that quercetin prevented biofilm production by 95% at 1/2 MIC (256 µg/mL).

Recent research has shown that quercetin has inhibitory effects against biofilm development in both reference and clinical isolates of *S. aureus* at sub-MICs. Biofilm generation by vancomycin-resistant *S. aureus* (VRSA) and methicillin-resistant *S. aureus* (MRSA) was reduced by almost half when quercetin was used (at doses ranging from 500 to 250 µg/mL). Quercetin significantly (*p* < 0.05) decreased biofilm generation after 24 h at both 20 µg/mL and 50 g/mL in a study that included the methicillin-resistant *S. aureus* strain (MSSA) ATCC 6538. It has been demonstrated that the anti-biofilm actions of flavonoids are affected by both the quantity and location of the hydroxyl group in their structures [31]. The flavonoid quercetin, which contains five hydroxyl groups, was found to reduce biofilm formation by more than 80%. Because of quercetin’s anti-virulence and anti-biofilm capabilities, it was also shown that red wines significantly increased the viability of *S. aureus* co-infected *Caenorhabditis elegans*.

The establishment of pathogen biofilm is dependent on QS [48]. Inhibiting the signal-mediated QS mechanism could thereby inhibit the growth of biofilms. At all tested concentrations, the results of this investigation show that quercetin significantly inhibited the biofilm growth of the tested microbes. Our results are consistent with those previously published [24,32], which stated that *S*. Typhimurium developed biofilms on food and food-contact surfaces when quercetin (125 µg/mL) treatment was used. According to a previous study [41], 0.2 mM of quercetin was employed to inhibit *L. monocytogenes* biofilms [24]. The quercetin reduced the biofilms of *L. monocytogenes*, indicating that it affects EPS production mechanisms [40]. However, at 0.2 and 0.4 mM, quercetin significantly (*p* < 0.05) increased biofilm inhibition by 1.96 and 3.21 Log10 CFU/cm^2^, respectively [41,42]. Quercetin’s effectiveness in preventing *S. epidermidis* biofilm development was examined [33]. At concentrations of 250 and 500 µg/mL, quercetin repressed the formation of *S. epidermidis* biofilm at 90.5 and 95.3%, respectively [34]. The actions of flavonoids, of which quercetin is a member, rely on the molecular structure of the compound. Flavonoids provide lipid protection through many pathways depending on its hydroxyl group arrangement, substitution, and total quantity. Specifically, quercetin can neutralize free radicals with a high oxidation potential. Quercetin’s ability to chelate metal ions is what allows it to suppress free radical production. Therefore, it is possible that their ability to inhibit bacterial adherence, enzymes, and disrupt membranes is related to their antimicrobial function.

Due to quercetin’s possible interruption of cell-to-cell connections, bacteria may lose their normal structure [1]. By severing these bonds, the biofilms of the cells are no longer bound together and can be washed away [32]. Inhibiting cell attachment is a key step in avoiding infections, and flavonoid drugs offer an auspicious approach for lowering bacterial colonization on surfaces and epithelial mucosa. Quercetin, a flavonoid, was found to impede biofilm formation by inhibiting cell adhesion, which is thought to alter the sticky nature of the bacterial cell wall. Quercetin breaks cell-to-cell interactions, as seen in the FE-SEM images (Figure 3), which is consistent with earlier results [24].

Numerous genetic factors are essential for the virulence and biofilm development of *L. monocytogenes*. In order to evaluate quercetin’s efficacy, we looked into the expression of *flaA*, *prfA*, *agrA*, *hlyA*, and *fbp* in *L. monocytogenes* cocktail culture strains. It is a new method for preventing the growth of biofilms, reducing the spread of pathogenic diseases, and safeguarding foodstuff. ROS accumulate inside the cell, which causes oxidative stress [49]. Oxidative stress has a major impact on biofilm development by improving microbial population adaptability and survival protection [36]. The antioxidant suppresses the development of biofilms and destabilizes the microbial cells’ membrane integrity [50]. In order to better understand the inhibitory mechanism, RT-PCR was used to assess the expression levels of the virulence and QS controller genes. The target genes were inhibited by using quercetin (Figure 4). The expression levels of *prfA, flgA, hlyA, fbp,* and *agrA* were suppressed by quercetin at 1/2 MIC when compared to control and 1/8 MIC (Figure 4). Since significant doses of quercetin are required to kill the bacterial cells, it is useful that a small amount of the compound can prevent biofilm development by controlling the expression of particular genes [1]. Additionally, quercetin, at a dose of 1/2 MIC, significantly suppressed the biofilm-associated genes *prfA, flgA, hlyA, fbp,* and *agrA* when compared to the other groups (Figure 4). According to earlier reports, quercetin’s effects on suppressing the virulence genes *actA, sigB, inlA, prfA*, and *inlC* were what caused the development of biofilm [5]. Inhibiting the production of the target genes suggests that quercetin lessens *L. monocytogenes*’ metabolic activity [5].

Quercetin enters cells and immediately affects intracellular goals or transcriptional regulators. Quercetin may interrelate with definite membrane proteins leading to transcriptional variations that have consequences for gene down-regulation. The polyhydroxy hydrolytic compound quercetin can form potent complexes with a variety of macromolecules (bacterial adhesins and cell membrane proteins). Through bacterial signaling mechanisms, such as two-component systems, membrane alterations may cause bacterial cells to adapt by altering the expressions of their genes.

## 5. Conclusions

Biofilm formation prevention is essential for ensuring food safety in the food processing business. *L. monocytogenes* biofilms on crabs and shrimps were shown to be inhibited by quercetin, and this is preliminary proof of the antibacterial effect of quercetin. The reduction in biofilms, using 1/2 MIC, on crabs and shrimps was 2.31 log and 2.36 log CFU/cm^2^, respectively (*p* < 0.05). Quercetin breaks cell-to-cell interactions, as evidenced by the FE-SEM images (Figure 3). Additionally, quercetin can repress virulence and QS genes. Quercetin, a plant-based nutritional component, is both inexpensive and efficient. Developing efficient control measures and employing appropriate methods for evaluating their efficacy is crucial in the battle against biofilms. The results of this research suggest that quercetin can be used as a biofilm inhibitor to lessen the prevalence of *L. monocytogenes* biofilms on foods (crabs and shrimps) in industrial kitchens and processing facilities.

## Figures and Tables

**Figure 1 antibiotics-12-01008-f001:**
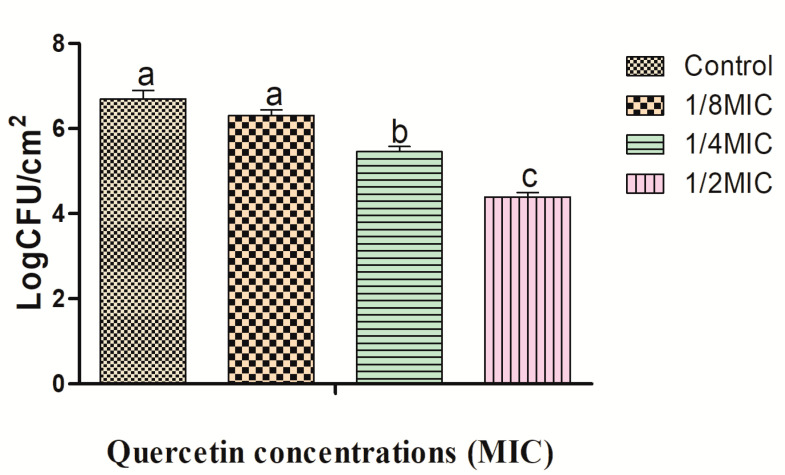
Biofilm formation of *Listeria monocytogenes* cocktail culture on crabs. All results are presented as mean ± SEM. Number of replicates (*n* = 3). Different letters (a–c) represent significant differences (*p* < 0.05).

**Figure 2 antibiotics-12-01008-f002:**
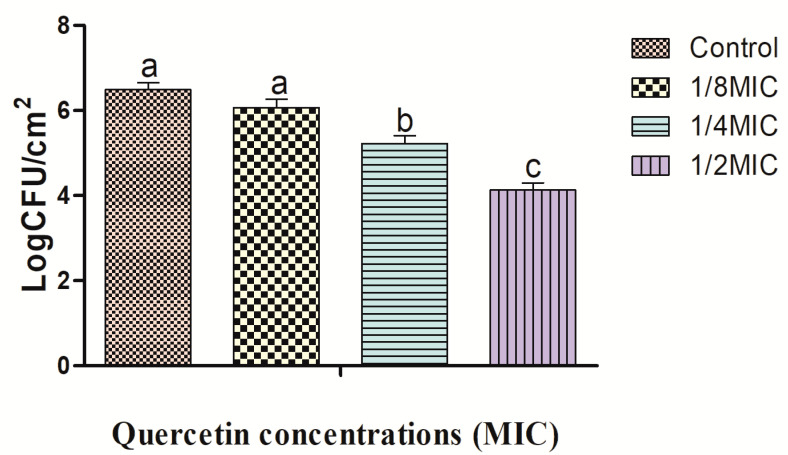
Biofilm formation of *Listeria monocytogenes* cocktail culture on shrimps. All results were presented as mean ± SEM. Number of replicates (*n* = 3). Different letters (a–c) represent significant differences (*p* < 0.05).

**Figure 3 antibiotics-12-01008-f003:**
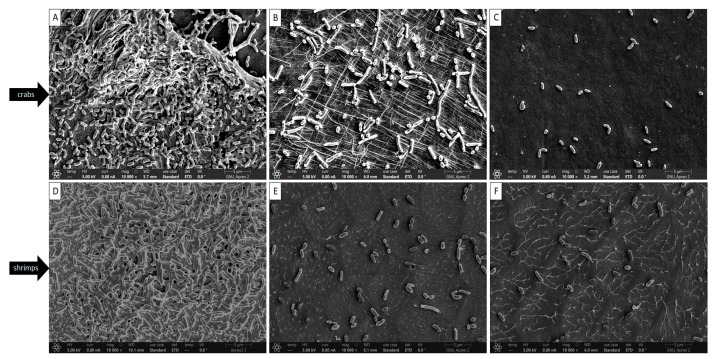
Graphical images of the inhibition of *Listeria monocytogenes* cocktail culture on the crabs: (**A**) control; (**B**) 1/4 MIC; (**C**) 1/2 MIC; and shrimps: (**D**) control; (**E**) 1/4 MIC; and (**F**) 1/2 MIC.

**Figure 4 antibiotics-12-01008-f004:**
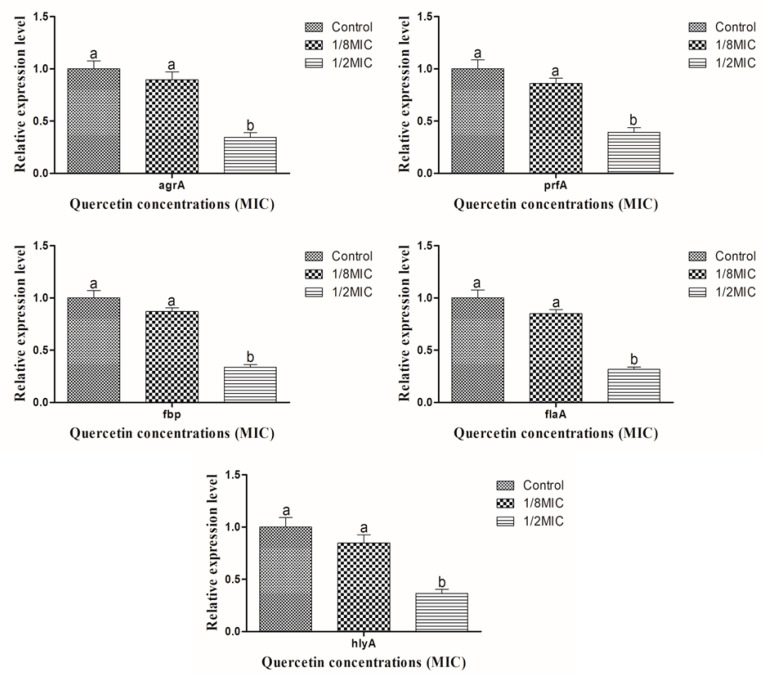
Suppression levels of *agrA*, *prfA*, *fbp*, *flaA*, and *hylA* genes in *Listeria monocytogenes* cocktail culture for control, 1/8, and 1/2 MIC quercetin. Number of replicates (*n* = 3). Different letters (a,b) represent significant differences (*p* < 0.05).

**Table 1 antibiotics-12-01008-t001:** List of primers for Real-Time PCR.

Primer Name	Primer Sequences (5′-3′)	Amplicon Size (bp)	Accession Number
*16S rRNA*	F: GGAGCATGTGGTTTAATTCG R: CCAACTAAATGCTGGCAACT	199	CP016470.1
*agrA*	F: ATGAAGCAAGCGGAAGAAC R: ACGACCTGTGACAACGATAAA	239	CP076669.1
*prfA*	F: CAATGGGATCCACAAGAATA R: AGCCTGCTCGCTAATGACTT	186	CP093220.1
*fbp*	F: GCCTGGTCTAAACTGGATTT R: CGCCATAAAGAGCGATACTT	189	CP090057.1
*flaA*	F: TGGTTCTACAGTTGCTGGTT R: TTTAGTTGCGATGGATTGGT	184	CP087264.1
*hlyA*	F: GCAATTTCGAGCCTAACCTA R: ACTGCGTTGTTAACGTTTGA	188	CP093220.1

## Data Availability

Data are contained within the article.

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
