# Peer review of "Inhibition of Listeria monocytogenes Cocktail Culture Biofilms on Crab and Shrimp Coupons and the Expression of Biofilm-Related Genes"

_antibiotics, 2023, doi:10.3390/antibiotics12061008_

Round 1

Reviewer 1 Report

Dear Editor and Authors

I started reading the paper

My first problem is the English, poor and in same cases I was not able to understand what the authors reported.

There are a lot of things not good and needing amendments.

I kindly ask to go checking the paper trying to amend all the chapters, based on my comments.

Some examples are:

There is chapter 3 with results and discussion and then chapter 4 with Discussion

At point 4 there is a lot of lines reporting information about different bacteria and not the results of your trial

Then I suggest the authors, first of all a deep revision of the paper taking into consideration what should be reported in the different paragraphs and then a grammar checking before a new submission, because based on me the paper should be rejected.

Dear Editor and Authors

I started reading the paper

My first problem is the English, poor and in same cases I was not able to understand what the authors reported.

There are a lot of things not good and needing amendments.

I kindly ask to go checking the paper trying to amend all the chapter, based on my comments.

Some examples are:

There is chapter 3 with results and discussion and then chapter 4 with Discussion

At point 4 there is a lot of lines reporting information about different bacteria and not the results of your trial

Then I suggest the authors, first of all a deep revision of the paper taking into consideration what should be reported in the different paragraphs and then a grammar checking before a new submission, because based on me the paper should be rejected.

Reviewer 2 Report

The manuscript looks at the effect of quercetin on biofilm formation on shrimp and crab coupons by a cocktail of three Listeria monocytogenes strains.

Abstract:

L17 Biofilm what? Formation?

L19 Add « log reduction of » after “as”.

L21 Poor structure, revisit. Idem L23, L124-125.

Introduction: too long with a lot of redundancy. Careful, like any chemical substance, polyphenol extracted for plants can be toxic/harmful/irritating at certain concentrations.

L30 Change “foodborne” for “ubiquitous”.

L52 Change “produce” for “products”.

L53-54 should follow L44-45 to avoid redundancy of topics.

L63 Add “are” before “used”.

L81 Change “is enable” for “enables”.

L97 Said already.

L104-105 Any reference on quercetin resistance ?

L107 Poor structure, revisit (Quercetin used traditionally to infectious disorders).

L112 A reference is missing after products.

L136 “genes” not “gene”.

L149-150 Poor structure, revisit (which was validated by on a PALCAM agar).

L155 “dissolved” not “desolving”

L180 Use exponent (cm2).

L183 Define abbreviation before using it: FE-SEM).

L201 Poor structure, revisit (After cultured the bacteria at 30 °C).

L208 20 L (liter) is a lot of Master Mix…

L211 Do not start a sentence with a number; this is basic scientific writing; also on L288.

L215 To confirm what?

L221 Poor structure, revisit (To analysis, significant difference (p < 0.05) analysis).

L230 A reference is missing after pathogenicity.

L241-243 Not cleat at all and lack a refence. Listeriosis without a capital letter.

L221 Poor structure, revisit (Different letters (a-c) are presented significant different (p < 0.05).)

L259 Add “that” after “reported”.

L260 A reference is missing after “study”.

Results and discussion or Results and Discussion separated. Titles do not work; see L225 and L280. Restructure the text accordingly.

Fig.4 Title of the figure should not be on the next page.

L283 “a” not “an” component. Compare the concentration with those used in the study.

L285 mg is an international unit no need to repeat in full.

L288 “content” is not the appropriate term.

L288-290 a sentence without a verb.

L293-294 Use and unbreakable space between “100” and “µg/mL”; also on L356 after “L.”

L305 A reference is missing after “structures”.

L308-309 Do the authors mean a co-infected biofilm; clarify.

L317 Which mechanisms?

L323-325 Redundancy with what was said before.

L330 Add communication after “cell-to-cell”.

L342-343 Poor structure, revisit (ROS buildup takes place privileged the cell, oxidative stress arises).

L345 Change “decreasing” by “destabilizes”.

L345 Add “used to” after “was”.

L349 Poor structure, revisit (by quercetin at sub-MICs (1/2 MIC) in compared to control and 1/8MIC (Figure 4)).

L358-359 Poor structure, revisit (straight influence intracellular objectives or transcrip-358 tional controllers.).

L360 Poor structure, revisit (that consequence in gene down-regulation.).

L363 Poor structure, revisit (by using signaling systems such two-component 363 systems to modify the expression of their genes).

Many suggestions have been made to improve the quality of the English language.

Round 2

Reviewer 2 Report

Introduction is the same length.

In the response to the review, the line numbers indicated does not correspond with the new version. This is not only annoying but shows a lack of thoroughness.

L100 what is fresh technology?

L198 add an unbreakable space after 24

L308 add an unbreakable space after S.
